# Network size and weights size for memorization with two-layers neural networks

**Sébastien Bubeck**
Microsoft Research

**Ronen Eldan** *
Weizmann Institute

**Yin Tat Lee**
University of Washington
& Microsoft Research

**Dan Mikulincer**
Weizmann Institute

## Abstract

In 1988, Eric B. Baum showed that two-layers neural networks with threshold activation function can perfectly memorize the binary labels of $n$ points in general position in $\mathbb{R}^d$ using only $\lceil n/d \rceil$ neurons. We observe that with ReLU networks, using four times as many neurons one can fit arbitrary real labels. Moreover, for approximate memorization up to error $\varepsilon$, the neural tangent kernel can also memorize with only $O\left(\frac{n}{d} \cdot \log(1/\varepsilon)\right)$ neurons (assuming that the data is well dispersed too). We show however that these constructions give rise to networks where the *magnitude* of the neurons' weights are far from optimal. In contrast we propose a new training procedure for ReLU networks, based on *complex* (as opposed to *real*) recombination of the neurons, for which we show approximate memorization with both $O\left(\frac{n}{d} \cdot \frac{\log(1/\varepsilon)}{\varepsilon}\right)$ neurons, as well as nearly-optimal size of the weights.

## 1 Introduction

We study two-layers neural networks in $\mathbb{R}^d$ with $k$ neurons and non-linearity $\psi : \mathbb{R} \to \mathbb{R}$. These are functions of the form:

$$x \mapsto \sum_{\ell=1}^{k} a_\ell \psi(w_\ell \cdot x + b_\ell), \tag{1}$$

with $a_\ell, b_\ell \in \mathbb{R}$ and $w_\ell \in \mathbb{R}^d$ for any $\ell \in [k]$. We are mostly concerned with the Rectified Linear Unit non-linearity, namely $\mathrm{ReLU}(t) = \max(0, t)$, in which case wlog one can restrict the recombination weights $(a_\ell)$ to be in $\{-1, 1\}$ (this holds more generally for positively homogeneous non-linearities). We denote by $\mathcal{F}_k(\psi)$ the set of functions of the form (1). Under mild conditions on $\psi$ (namely that it is not a polynomial), such neural networks are *universal*, in the sense that for $k$ large enough they can approximate any continuous function [Cybenko, 1989, Leshno et al., 1993].

In this paper we are interested in approximating a target function on a *finite* data set. This is also called the *memorization* problem. Specifically, fix a data set $(x_i, y_i)_{i \in [n]} \in (\mathbb{R}^d \times \mathbb{R})^n$ and an approximation error $\varepsilon > 0$. We denote $\mathbf{y} = (y_1, \ldots, y_n)$, and for a function $f : \mathbb{R}^d \to \mathbb{R}$ we write $\mathbf{f} = (f(x_1), \ldots, f(x_n))$. The main question concerning the memorization capabilities of $\mathcal{F}_k(\psi)$ is as follows: How large should be $k$ so that there exists $f \in \mathcal{F}_k(\psi)$ such that $\|\mathbf{f} - \mathbf{y}\|^2 \leq \varepsilon \|\mathbf{y}\|^2$ (where $\|\cdot\|$ denotes the Euclidean norm)? A simple consequence of universality of neural networks is that $k \geq n$ is sufficient (see Proposition 2). In fact (as was already observed by Baum [1988] for threshold $\psi$ and binary labels, see Proposition 3) much more compact representations can be achieved by leveraging the high-dimensionality of the data. Namely we prove that for $\psi = \mathrm{ReLU}$ and a data set in general position (i.e., any hyperplane contains at most $d$ points), one only needs $k \geq 4 \cdot \lceil \frac{n}{d} \rceil$ to memorize the data perfectly, see Proposition 4. The size $k \approx n/d$ is clearly optimal,

by a simple parameter counting argument. We call the construction given in Proposition 4 a *Baum network*, and as we shall see it is of a certain combinatorial flavor. In addition we also prove that such memorization can in fact essentially be achieved in a kernel regime (with a bit more assumptions on the data): we prove in Theorem 2 that for $k = \Omega\left(\frac{n}{d}\log(1/\varepsilon)\right)$ one can obtain approximate memorization with the Neural Tangent Kernel [Jacot et al., 2018], and we call the corresponding construction the *NTK network*. Specifically, the kernel we consider is,

$$\mathbb{E}\left[\nabla_w \psi(w \cdot x) \cdot \nabla_w \psi(w \cdot y)\right] = \mathbb{E}\left[(x \cdot y)\psi'(w \cdot x)\psi'(w \cdot y)\right],$$

where $\nabla_w$ is the gradient with respect to the $w$ variable and the expectation is taken over a random initialization of $w$.

**Measuring regularity via total weight.** One is often interested in fitting the data using functions which satisfy certain regularity properties. The main notion of regularity in which we are interested is the *total weight*, defined as follows: For a function $f : \mathbb{R}^d \to \mathbb{R}$ of the form (1), we define

$$\mathbf{W}(f) := \sum_{\ell=1}^{k} |a_\ell|\sqrt{\|w_\ell\|^2 + b_\ell^2}.$$

This definition is widely used in the literature, see Section 2 for a discussion and references. Notably, it was shown in Bartlett [1998] that this measure of complexity is better associated with the network's generalization ability compared to the size of the network. We will be interested in constructions which have both a small number of neurons and a small total weight.

**Our main contribution: The complex network.** As we will see below, both the Baum network and the NTK networks have sub-optimal total weight. The main technical contribution of our paper is a third type of construction, which we call the *harmonic network*, that under the same assumptions on the data as for the NTK network, has both near-optimal memorization size and near-optimal total weight:

**Theorem 1** *(Informal). Suppose that $n \le \mathrm{poly}(d)$. Let $x_1, .., x_N \in \mathbb{S}^{d-1}$ such that*

$$|x_i \cdot x_j| = \widetilde{O}\left(\frac{1}{\sqrt{d}}\right).$$

*For every $\varepsilon > 0$ and every choice of labels $(y_i)_{i=1}^n$ such that $|y_i| = O(1)$ for all $i$, there exist $k = \widetilde{O}\left(\frac{n}{d\varepsilon}\right)$ and $f \in \mathcal{F}_k(\psi)$ such that*

$$\frac{1}{n}\sum_{i=1}^{n} \min\left(\left(y_i - f(x_i)\right)^2, 1\right) \le \varepsilon$$

*and such that $\mathbf{W}(f) = \widetilde{O}\left(\sqrt{n}\right)$.*

We show below in Proposition 1 that for random data one necessarily has $\mathbf{W}(f) = \widetilde{\Omega}\left(\sqrt{n}\right)$, thus proving that the harmonic network has near-optimal total weight. Moreover we also argue in the corresponding sections that the Baum and NTK networks have total weight at least $n\sqrt{n}$ on random data, thus being far from optimal.

**An iterative construction.** Both the NTK network and the harmonic network will be built by iteratively adding up small numbers of neurons. This procedure, akin to boosting, is justified by the following lemma. It shows that to build a large memorizing network it suffices to be able to build a small network $f$ whose scalar product with the data $\mathbf{f} \cdot \mathbf{y}$ is comparable to its variance $\|\mathbf{f}\|^2$:

**Lemma 1** *Fix $(x_i)_{i=1}^n$. Suppose that there are $m \in \mathbb{N}$ and $\alpha, \beta > 0$ such that the following holds: For any choice of $(y_i)_{i=1}^n$, there exists $f \in \mathcal{F}_m(\psi)$ with $\mathbf{y} \cdot \mathbf{f} \ge \alpha\|\mathbf{y}\|^2$ and $\|\mathbf{f}\|^2 \le \beta\|\mathbf{y}\|^2$. Then for all $\varepsilon > 0$, there exists $g \in \mathcal{F}_{mk}(\psi)$ such that*

$$\|\mathbf{g} - \mathbf{y}\|^2 \le \varepsilon\|\mathbf{y}\|^2$$

*with*

$$k \le \frac{\beta}{\alpha^2}\log(1/\varepsilon).$$

*Moreover, if the above holds with $\mathbf{W}(f) \le \omega$, then $\mathbf{W}(g) \le \frac{\omega}{\alpha}\log(1/\varepsilon)$.*

**Proof.** Denote $\eta = \frac{\alpha}{\beta}$ and $\mathbf{r}_1 = \mathbf{y}$. Then, there exists $f_1 \in \mathcal{F}_m(\psi)$, such that

$$\|\eta \mathbf{f}_1 - \mathbf{r}_1\|^2 = \|\mathbf{r}_1\|^2 - 2\eta \mathbf{y} \cdot \mathbf{f}_1 + \eta^2 \|\mathbf{f}_1\|^2 \leq \|\mathbf{r}_1\|^2 \left(1 - 2\frac{\alpha^2}{\beta} + \frac{\alpha^2}{\beta}\right)$$

$$\leq \|\mathbf{r}_1\|^2 \left(1 - \frac{\alpha^2}{\beta}\right) = \|\mathbf{y}\|^2 \left(1 - \frac{\alpha^2}{\beta}\right)$$

The result is obtained by iterating the above inequality with $\mathbf{r}_i = \mathbf{y} - \eta \sum_{j=1}^{i-1} \mathbf{f}_j$ taken as the residuals. By induction, if we set $g = \eta \sum_{j=1}^{k} f_j$, we get

$$\|\mathbf{g} - \mathbf{y}\| = \|\eta \mathbf{f}_k - \mathbf{r}_k\| \leq \|\mathbf{r}_k\|^2 \left(1 - \frac{\alpha^2}{\beta}\right) = \|\mathbf{y}\|^2 \left(1 - \frac{\alpha^2}{\beta}\right)^k.$$

$\square$

In both the NTK and harmonic constructions, the function $f$ will have the largest possible correlation with the data set attainable for a network of constant size. However, the harmonic network will have the extra advantage that the function $f$ will be composed of a single neuron whose weight is the smallest one attainable. Thus, the harmonic network will enjoy both the smallest possible number of neurons and smallest possible total weight (up to logarithmic factors). Note however that the dependency on $\varepsilon$ is worse for the harmonic network, which is technically due to a constant order term in the variance which we do not know how to remove.

We conclude the introduction by showing that a total weight of $\Omega(\sqrt{n})$ is necessary for approximate memorization. Just like for the upper bound, it turns out that it is sufficient to consider how well can one correlate a single neuron. Namely the proof boils down to showing that a single neuron cannot correlate well with random data sets.

**Proposition 1** *There exists a data set $(x_i, y_i)_{i \in [n]} \in (\mathbb{S}^{d-1} \times \{-1, 1\})^n$ such that for every function $f$ of the form* (1) *with $\psi$ $L$-Lipschitz and which satisfies $\|\mathbf{f} - \mathbf{y}\|^2 \leq \frac{1}{2}\|\mathbf{y}\|^2$, it holds that $\mathbf{W}(f) \geq \frac{\sqrt{n}}{8L}$.*

**Proof.** We have

$$\frac{1}{2}\|\mathbf{y}\|^2 \geq \|\mathbf{f} - \mathbf{y}\|^2 \geq \|\mathbf{y}\|^2 - 2\mathbf{f} \cdot \mathbf{y} \Rightarrow \mathbf{f} \cdot \mathbf{y} \geq \frac{1}{4}\|\mathbf{y}\|^2 \,,$$

that is

$$\sum_{\ell=1}^{k} \sum_{i=1}^{n} y_i a_\ell \psi(w_\ell \cdot x_i - b_\ell) \geq \frac{n}{4} \,,$$

which implies:

$$\max_{w,b} \sum_{i=1}^{n} y_i \frac{\psi(w \cdot x_i - b)}{\sqrt{\|w\|^2 + b^2}} \geq \frac{n}{4\mathbf{W}(f)} \,.$$

Now let us assume that $y_i$ are $\pm 1$ uniformly at random (i.e., Rademacher random variables), and thus by Talagrand's contraction lemma for the Rademacher complexity (see [Lemma 26.9, Shalev-Shwartz and Ben-David [2014]]) we have:

$$\mathbb{E} \max_{w,b} \sum_{i=1}^{n} y_i \frac{\psi(w \cdot x_i - b)}{\sqrt{\|w\|^2 + b^2}} \leq L \cdot \mathbb{E} \max_{w,b} \sum_{i=1}^{n} y_i \frac{w \cdot x_i - b}{\sqrt{\|w\|^2 + b^2}}$$

$$\leq L \cdot \mathbb{E} \sqrt{\left\|\sum_{i=1}^{n} y_i x_i\right\|^2 + n} \leq 2L\sqrt{n} \,,$$

and thus $\mathbf{W}(f) \geq \frac{\sqrt{n}}{8L}$.

$\square$

## 2 Related works

**Exact memorization.**  The observation that $n$ neurons are sufficient for memorization with essentially arbitrary non-linearity was already made in [Bach, 2017] (using Carathéodory's theorem), and before that a slightly weaker bound with $n+1$ neurons was already observed in [Bengio et al., 2006] (or more recently $2n + d$ in [Zhang et al., 2017]). The contribution of Proposition 2 is to show that this statement of exactly $n$ neurons follows in fact from elementary linear algebra.

As already mentioned above, Baum [1988] proved that for threshold non-linearity and binary labels one can obtain a much better bound of $n/d$ neurons for memorization, as long as the data is in general position. This was generalized to the ReLU non-linearity (but still binary labels) in Yun et al. [2019] (we note that this paper also considers some questions around memorization capabilities of deeper networks). Our modest contribution here is to generalize this to arbitrary real labels, see Proposition 4.

**Gradient-based memorization.**  A different line of works on memorization studies whether it can be achieved via gradient-based optimization on various neural network architectures. The literature here is very large, but early results with minimal assumptions include Li and Liang [2018], Soltanolkotabi et al. [2018] which were notably generalized in [Allen-Zhu et al., 2019, Du et al., 2019]. Crucially these works leverage very large overparametrization, i.e., the number of neurons is a large polynomial in the number of data points. For a critique of this large overparametrization regime see [Chizat et al., 2019, Ghorbani et al., 2019, Yehudai and Shamir, 2019], and for a different approach based on a certain scaling limit of stochastic gradient descent for sufficiently overparametrized networks see [Chizat and Bach, 2018, Mei et al., 2018]. More recently the amount of overparametrization needed was improved to a small polynomial dependency in $n$ and $d$ in [Kawaguchi and Huang, 2019, Oymak and Soltanolkotabi, 2019, Song and Yang, 2019]. In the *random features* regime, Bresler and Nagaraj [2020] have also considered an iterative construction procedure for memorization. This is somewhat different than our approach, in which the iterative procedure updates the $w_j$'s, and a much smaller number of neurons is needed as a result. Finally, very recently Amit Daniely [Daniely, 2019, 2020] showed that gradient descent already works in the optimal regime of $k = \widetilde{O}(n/d)$, at least for random data (and random labels). This result is closely related to our analysis of the NTK network in Section 4. Minor distinctions are that we allow for arbitrary labels, and we take a "boosting approach" were neurons are added one by one (although we do not believe that this is an essential difference).

**Total weight complexity.**  It is well-known since Bartlett [1998] that the total weight of a two-layers neural network is a finer measure of complexity than the number of neurons to control its generalization (see Neyshabur et al. [2015] and Arora et al. [2019] for more recent discussions on this, as well as Bartlett et al. [2017] for other notions of norms for deeper networks). Of course the bound $\mathbf{W} = \widetilde{O}(\sqrt{n})$ proved here leads to vacuous generalization performance, as is necessary since the Harmonic network can memorize completely random data (for which no generalization is possible). It would be interesting to see if the weight of the Harmonic network can be smaller for more structured data, particularly given the context raised by the work [Zhang et al., 2017] (where it was observed that SGD on deep networks will memorize arbitrary data, hence the question of where does the seeming generalization capabilities of those networks come from). We note the recent work [Ji and Telgarsky, 2020] which proves for example that polylogarithmic size network is possible for memorization under a certain margin condition. Finally we also note that the effect in function space of bounding $\mathbf{W}$ has been recently studied in Ongie et al. [2020], Savarese et al. [2019].

**Complex weights.**  It is quite natural to consider neural networks with complex weights. Indeed, as was already observed by Barron [Barron, 1993], the Fourier transform $f(x) = \int \hat{f}(\omega) \exp(i\omega \cdot x) d\omega$ exactly gives a representation of $f$ as a two-layers neural network with the non-linearity $\psi(t) = \exp(it)$. More recently, it was noted in Andoni et al. [2014] that randomly perturbing a neuron with *complex weights* is potentially more beneficial than doing a mere real perturbation. We make a similar observation in Section 5 for the construction of the Harmonic network, where we show that complex perturbations allow to deal particularly easily with higher order terms in some key Taylor expansion. Moreover we also note that Andoni et al. [2014] considers non-linearity built from Hermite polynomials, which shall be a key step for us too in the construction of the Harmonic

network (the use of Hermite polynomials in the context of learning theory goes back to [Kalai et al., 2008]).

While orthogonal to our considerations here, we also note the work of Fefferman [Fefferman, 1994], where he used the analytical continuation of a (real) neural network to prove a certain uniqueness property (essentially that two networks with the same output must have the same weights up to some obvious symmetries and obvious counter-examples).

## 3   Elementary results on memorization

In this section we give a few examples of elementary conditions on $k$, $\psi$ and the data set so that one can find $f \in \mathcal{F}_k(\psi)$ with $\mathbf{f} = \mathbf{y}$ (i.e., exact memorization). We prove three results: (i) $k \geq n$ suffices for any non-polynomial $\psi$, (ii) $k \geq \frac{n}{d} + 3$ with $\psi(t) = \mathbb{1}\{t \geq 0\}$ suffices for binary labels with data in general position (this is exactly Baum [1988]'s result), and (iii) $k \geq 4 \cdot \lceil \frac{n}{d} \rceil$ with $\psi = \mathrm{ReLU}$ suffices for data in general position and arbitrary labels.

We start with the basic linear algebraic observation that having a number of neurons larger than the size of the data set is always sufficient for perfect memorization:

**Proposition 2** *Assuming that $\psi$ is not a polynomial, there exists $f \in \mathcal{F}_n(\psi)$ such that $\mathbf{f} = \mathbf{y}$.*

**Proof.** Note that the set of functions of the form (1) (with arbitrary $k$) corresponds to the vector space $V$ spanned by the functions $\psi_{w,b} : x \mapsto \psi(w \cdot x + b)$. Consider the linear operator $\Psi : V \to \mathbb{R}^n$ that corresponds to the evaluation on the data points $(x_i)$ (i.e., $\Psi(f) = (f(x_i))_{i \in [n]}$). Since $\psi$ is not a polynomial, the image of $\Psi$ is $\mathrm{Im}(\Psi) = \mathbb{R}^n$. Moreover $\mathrm{Im}(\Psi)$ is spanned by the set of vectors $\Psi(\psi_{w,b})$ for $w \in \mathbb{R}^d, b \in \mathbb{R}$. Now, since $\dim(\mathrm{Im}(\Psi)) = n$, one can extract a subset of $n$ such vectors with the same span, that is there exists $w_1, b_1, \ldots, w_n, b_n$ such that

$$\mathrm{span}(\Psi(\psi_{w_1,b_1}), \ldots, \Psi(\psi_{w_n,b_n})) = \mathbb{R}^n \,,$$

which concludes the proof.                                                                                $\square$

In [Baum, 1988] it is observed that one can dramatically reduce the number of neurons for high-dimensional data:

**Proposition 3** *Fix $\psi(t) = \mathbb{1}\{t \geq 0\}$. Let $(x_i)_{i \in [n]}$ be in general position in $\mathbb{R}^d$ (i.e., any hyperplane contains at most $d$ points), and assume binary labels, i.e., $y_i \in \{0, 1\}$. Then there exists $f \in \mathcal{F}_{\frac{n}{d}+3}(\psi)$ such that $\mathbf{f} = \mathbf{y}$.*

**Proof.** Baum [1988] builds a network iteratively as follows. Pick $d$ points with label 1, say $x_1, \ldots, x_d$, and let $H = \{x : u \cdot x = b\}$ be a hyperplane containing those points and no other points in the data, i.e., $x_i \notin H$ for any $i > d$. With two neurons (i.e., $f \in \mathcal{F}_2(\psi)$) one can build the indicator of a small neighborhood of $H$, namely $f(x) = \psi(u \cdot x - (b - \tau)) - \psi(u \cdot x - (b + \tau))$ with $\tau$ small enough, so that $f(x_i) = 1$ for $i \leq d$ and $f(x_i) = 0$ for $i > d$. Assuming that the label 1 is the minority (which is without loss of generality up to one additional neuron), one thus needs at most $2 \lceil \frac{n}{2d} \rceil$ neurons to perfectly memorize the data.                                                                                $\square$

We now extend Proposition 3 to the ReLU non-linearity and arbitrary real labels. To do so we introduce the *derivative neuron* of $\psi$ defined by:

$$f_{\delta,u,v,b} : x \mapsto \frac{\psi((u + \delta v) \cdot x - b) - \psi(u \cdot x - b)}{\delta} \,, \tag{2}$$

with $\delta \in \mathbb{R}$ and $u, v \in \mathbb{R}^d$. As $\delta$ tends to 0, this function is equal to

$$f_{u,v,b}(x) = \psi'(u \cdot x - b)v \cdot x \tag{3}$$

for any $x$ such that $\psi$ is differentiable at $u \cdot x - b$. In fact, for the ReLU one has for any $x$ such that $u \cdot x \neq b$ that $f_{\delta,u,v,b}(x) = f_{u,v,b}(x)$ for $\delta$ small enough (this is because the ReLU is piecewise

linear). We will always take $\delta$ small enough and $u$ such that $f_{\delta,u,v,b}(x_i) = f_{u,v,b}(x_i)$ for any $i \in [n]$, for example by taking

$$\delta = \frac{1}{2} \min_{i \in [n]} \frac{|u \cdot x_i - b|}{|v \cdot x_i|} . \qquad (4)$$

Thus, as far as memorization is concerned, we can assume that $f_{u,v,b} \in \mathcal{F}_2(\text{ReLU})$. With this observation it is now trivial to prove the following extension of Baum's result:

**Proposition 4** *Let $(x_i)_{i \in [n]}$ be in general position in $\mathbb{R}^d$ (i.e., any hyperplane contains at most $d$ points). Then there exists $f \in \mathcal{F}_{4 \cdot \lceil \frac{n}{d} \rceil}(\text{ReLU})$ such that $\mathbf{f} = \mathbf{y}$.*

**Proof.** Pick an arbitrary set of $d$ points, say $(x_i)_{i \leq d}$, and let $H = \{x : u \cdot x = b\}$ be a hyperplane containing those points and no other points in the data, i.e., $x_i \notin H$ for any $i > d$. With four neurons one can build the function $f = f_{u,v,b-\tau} - f_{u,v,b+\tau}$ with $\tau$ small enough so that $f(x_i) = x_i \cdot v$ for $i \leq d$ and $f(x_i) = 0$ for $i > d$. It only remains to pick $v$ such that $v \cdot x_i = y_i$ for any $i \leq d$, which we can do since the matrix given by $(x_i)_{i \leq d}$ is full rank (by the general position assumption). $\qquad \square$

## 4 The NTK network

The constructions in Section 3 are based on a very careful set of weights that depend on the entire dataset. Here we show that essentially the same results can be obtained in the *neural tangent kernel* regime. That is, we take pair of neurons as given in (2) (which corresponds in fact to (3) since we will take $\delta$ to be small, we will also restrict to $b = 0$), and crucially we will also have that the "main weight" $u$ will be chosen at random from a standard Gaussian, and only the "small perturbation" $v$ will be chosen as a function of the dataset. The guarantee we obtain is slightly weaker than in Proposition 4: we have a $\log(1/\varepsilon)$ overhead in the number of neurons, and moreover we also need to assume that the data is "well-spread". Specifically we consider the following notion of "generic data":

**Definition 1** *We say that $(x_i)_{i \in [n]}$ are $(\gamma, \omega)$-generic (with $\gamma \in (\frac{1}{2n}, 1)$ and $\omega > 0$) if $\|x_i\| \geq 1$ for all $i \in [n]$, $\frac{1}{n} \sum_{i=1}^{n} x_i x_i^\top \preceq \frac{\omega}{d} \cdot \mathrm{I}_d$, and $|x_i \cdot x_j| \leq \gamma \cdot \|x_i\| \cdot \|x_j\|$ for all $i \neq j$.*

In the following we fix such a $(\gamma, \omega)$-generic data set. Note that i.i.d. points on the sphere are $\left( O\left( \sqrt{\frac{\log(n)}{d}} \right), O(1) \right)$-generic. We now formulate our main theorem concerning the NTK network.

**Theorem 2** *There exists $f \in \mathcal{F}_k(\text{ReLU})$, produced in the NTK regime (see Theorem 3 below for more details) with $\mathbb{E}[\|\mathbf{f} - \mathbf{y}\|^2] \leq \varepsilon \|\mathbf{y}\|^2$ (the expectation is over the random initialization of the "main weights") provided that*

$$k \cdot d \geq 20\omega \cdot n \log(1/\varepsilon) \cdot \frac{\log(2n)}{\log(1/\gamma)} . \qquad (5)$$

In light of Lemma 1, it will be enough to produce a width-2 network, $f \in \mathcal{F}_2(\text{ReLU})$, whose correlation with the data set is large.

**Theorem 3** *There exists $f \in \mathcal{F}_2(\text{ReLU})$ with*

$$\mathbf{y} \cdot \mathbf{f} \geq \frac{1}{10} \cdot \sqrt{\frac{\log(1/\gamma)}{\log(2n)}} \cdot \|\mathbf{y}\|^2 , \qquad (6)$$

*and*

$$\|\mathbf{f}\|^2 \leq \frac{\omega \cdot n}{d} \|\mathbf{y}\|^2 . \qquad (7)$$

*In fact, one can take the construction (2) with:*

$$u \sim \mathcal{N}(0, \mathrm{I}_d), \ v = \sum_{i:u \cdot x_i \geq 0} y_i x_i, \ \delta = \frac{1}{2} \frac{\min_{i \in [n]} |u \cdot x_i|}{|v \cdot x_i|}. \qquad (8)$$

*which produces $f \in \mathcal{F}_2(\text{ReLU})$ such that (6) holds in expectation and (7) holds almost surely.*

To deduce Theorem 2 from Theorem 3, apply Lemma 1 with $\alpha = \frac{1}{10} \cdot \sqrt{\frac{\log(1/\gamma)}{\log(2n)}}$ and $\beta = \frac{\omega \cdot n}{d}$. See supplementary material for the proof of Theorem 3.

For $u \in \mathbb{R}^d$, set

$$f_u(x) = \psi'(u \cdot x) v \cdot x, \tag{9}$$

where $v$ is defined as in (8). Observe that as long as $u \cdot x_i \neq 0, \forall i \in [n]$, a small enough choice of $\delta$ ensures the existence of $f \in \mathcal{F}_2(\mathrm{ReLU})$ such that $\mathbf{f} = \mathbf{f}_u$.

To prove Theorem 3, it therefore remains to show that $\mathbf{f}_u$ satisfies (6) and (7) with positive probability as $u \sim \mathcal{N}(0, \mathrm{I}_d)$. This will be carried out in two steps: First we show that the correlation $\mathbf{y} \cdot \mathbf{f}$ for a derivative neuron has a particularly nice form as a function of $u$, see Lemma 2. Then, in Lemma 3 we derive a lower bound for the expectation of the correlation under $u \sim \mathcal{N}(0, \mathrm{I}_d)$. Taken together these lemmas complete the proof of Theorem 3.

**Lemma 2** *Fix* $u \in \mathbb{R}^d$, *the function* $f_u$ *defined in* (9) *satisfies* $\sum_{i=1}^n y_i f_u(x_i) = \left\| \sum_{i: u \cdot x_i \geq 0} y_i x_i \right\|^2$, *and furthermore* $\sum_{i=1}^n f_u(x_i)^2 \leq \frac{\omega \cdot n}{d} \cdot \sum_{i=1}^n y_i f(x_i)$.

**Proof.** We may write

$$\sum_{i=1}^n f_u(x) y_i = \sum_{i=1}^n \psi'(u \cdot x_i) y_i x_i \cdot v \,.$$

To maximize this quantity we take $v = \sum_{i=1}^n \psi'(u \cdot x_i) y_i x_i$ so that the correlation is exactly equal to $\|v\|^2 = \|\sum_{i=1}^n \psi'(u \cdot x_i) y_i x_i\|^2$ (note also that $\psi'(t) = \mathbb{1}\{t \geq 0\}$ for the ReLU). Moreover we also have (recall that for ReLU, $|\psi'(t)| \leq 1$)

$$\sum_{i=1}^n f_u(x_i)^2 = \sum_{i=1}^n (\psi'(x_i \cdot u))^2 (x_i \cdot v)^2 \leq \lambda_{\max}\left(\sum_{i=1}^n x_i x_i^\top\right) \cdot \|v\|^2 \,. \tag{10}$$

$\square$

**Lemma 3** *One has:*

$$\mathbb{E}_{u \sim \mathcal{N}(0, \mathrm{I}_n)} \left\| \sum_{i: \, u \cdot x_i \geq 0} y_i x_i \right\|^2 \geq \frac{1}{10} \cdot \sqrt{\frac{\log(1/\gamma)}{\log(2n)}} \cdot \sum_{i=1}^n y_i^2 \|x_i\|^2 \,.$$

**Proof.** First note that $\mathbb{E} \left\| \sum_{i: \, u \cdot x_i \geq 0} y_i x_i \right\|^2 = \mathbf{y}^\top H \mathbf{y}$, where

$$H_{i,j} = \mathbb{E}[x_i \cdot x_j \mathbb{1}\{u \cdot x_i \geq 0\} \mathbb{1}\{u \cdot x_j \geq 0\}] = \frac{2}{\pi} x_i \cdot x_j \left( \frac{1}{4} + \arcsin\left( \frac{x_i}{\|x_i\|} \cdot \frac{x_j}{\|x_j\|} \right) \right) \,.$$

Let us denote $V$ the matrix with entries $V_{i,j} = \frac{x_i}{\|x_i\|} \cdot \frac{x_j}{\|x_j\|}$ and $D$ the diagonal matrix with entries $\|x_i\|$. Note that $V \succeq 0$ and thus we have (recall also that $\arcsin(t) = \sum_{i=0}^\infty \frac{(2i)!}{(2^i i!)^2} \cdot \frac{t^{2i+1}}{2i+1}$):

$$D^{-1} H D^{-1} \succeq \frac{2}{\pi} \sum_{i=0}^\infty \frac{(2i)!}{(2^i i!)^2} \cdot \frac{V^{\circ 2(i+1)}}{2i+1} \,.$$

Now observe that for any $i$, by the Schur product theorem one has $V^{\circ i} \succeq 0$. Moreover $V^{\circ i}$ is equal to $1$ on the diagonal, and off-diagonal it is smaller than $\gamma^i$, and thus for $i \geq \frac{\log(2n)}{\log(1/\gamma)}$ one has $V^{\circ i} \succeq \frac{1}{2} \mathrm{I}_n$. The conclusion now follows by an easy calculation. $\square$

## 5 The complex network

We now wish to improve upon the NTK construction, by creating a network with similar memorization properties and which has almost no excess total weight.

## 5.1 Correlation of a perturbed neuron with random sign

Towards understanding our construction, let us first revisit the task of correlating a *single* neuron with the data, namely we want to maximize over $w$ the ratio between $|\sum_{i=1}^{n} y_i \psi(w \cdot x_i)|$ and $\sqrt{\sum_{i=1}^{n} \psi(w \cdot x_i)^2}$. Note that depending on whether the sign of the correlation is positive or negative, one would eventually take either neuron $x \mapsto \psi(w \cdot x)$ or $x \mapsto -\psi(w \cdot x)$. Let us first revisit the NTK calculation from the previous section, emphasizing that one can take a random sign for the recombination weight $a$.

The key NTK-like observation is that a single neuron perturbed around the parameter $w_0$ and with random sign can be interpreted as a linear model over a feature mapping that depends on $w$. More precisely (note that the random sign cancels the $0^{th}$ order term in the Taylor expansion):

$$\mathbb{E}_{a \sim \{-\delta,\delta\}} \, a^{-1} \psi\big((w + av) \cdot x\big) = \Phi_w(x) \cdot v + O(\delta) \, , \text{ where } \Phi_w(x) = \psi'(w \cdot x)x \, . \quad (11)$$

In particular the correlation to the data of such a single random neuron is equal in expectation to $\sum_i y_i \Phi_w(x_i) \cdot v + O(\delta)$, and thus it is natural to take the perturbation vector $v$ to be equal to $v_0 = \eta \sum_i y_i \Phi_w(x_i)$ (where $\eta$ will be optimized to balance with the variance term), and we now find that:

$$\mathbb{E}_{a \sim \{-\delta,\delta\}} \sum_{i=1}^{n} y_i a^{-1} \psi((w + av_0) \cdot x_i) = \left\| \eta \sum_i y_i \Phi_w(x_i) \right\|^2 + O(\delta) = \eta y^\top H(w) y + O(\delta) \, , \quad (12)$$

where $H(w)$ is the Gram matrix of the feature embedding, namely

$$H(w)_{i,j} = \Phi_w(x_i) \cdot \Phi_w(x_j).$$

Note that for $\psi = ReLU$, one has in fact that the term $O(\delta)$ in (11) disappears for $\delta$ small is enough, and thus the correlation to the data is simply $\eta y^\top H(w) y$ in that case.

As we did with the NTK network, we now also take the base parameter $w$ at random from a standard Gaussian. As we just saw, understanding the expected correlation then reduces to lower bound (spectrally) the Gram matrix $H$ defined by $H_{i,j} = \mathbb{E}_{w \sim \mathcal{N}(0, \mathrm{I}_d)}[\psi'(w \cdot x_i)\psi'(w \cdot x_j)x_i \cdot x_j]$. This was exactly the content of Lemma 3 for $\psi = \mathrm{ReLU}$.

## 5.2 Eliminating the higher derivatives with a complex trick

The main issue of the strategy described above is that it requires to take $\delta$ small, which in turn may significantly increase the total weights of the resulting network. Our next idea is based on the following observation: Taking a random sign in (11) eliminates all the even order term in the Taylor expansion since $\mathbb{E}_{a \sim \{-1,1\}}[a^{-1}a^m] = 0$ for any even $m$ (while it is $= 1$ for any odd $m$). However, taking a *complex* $a$, would rid us of *all* terms except the first order term. Namely, one has $\mathbb{E}_{a \in \mathbb{C}: |a|=1}[a^{-1}a^m] = 0$ for any $m \neq 1$. This suggests that it might make sense to consider neurons of the form

$$x \mapsto \mathrm{Re}\left(a^{-1}\psi\big((w + av) \cdot x\big)\right),$$

where $a$ is a complex number of unit norm.

The challenge is now to give sense to $\psi(z)$ for a complex $z$, so that the rest of the argument remains unchanged. This gives rise to two caveats:

- There is no holomorphic extension of the $\mathrm{ReLU}$ function.
- The holomorphic extension of the activation function, even if exists, is a function of two (real) variables. The expression $\psi\big((w + av) \cdot x\big)$ when $a \notin \mathbb{R}$ is not a valid neuron to be used in our construction since we're only allowed to use the original activation function as our non-linearity.

To overcome these caveats, the construction will be carried out in two steps, where in the first step we use *polynomial* activation functions, and in the second step, we replace these by the original activation function. It turns out that the calculation in Lemma 3 is particularly simple when the

derivative of the activation function is a Hermite polynomial (see supplementary material for definitions), which is in particular obviously well-defined on $\mathbb{C}$ and in fact holomorphic.

The first step of our proof in the supplementary material will be to obtain a result analogous to Theorem 1 where the ReLU is replaced by a Hermite activation. Given such a result, the second step towards Theorem 1 is to replace the polynomial attained by the above lemma by a ReLU. This will be achieved by:

- Observing that any polynomial in two variables $p(x, y)$ can be written as a linear combination of polynomials which only depend on one direction, hence polynomials of the form $q(ax + by)$.

- Using the fact that any nice enough function of one variable can be written as a mixture of ReLUs, due to the fact that the second derivative of the ReLU is a Dirac function (this was observed before, see e.g., [Lemma A.4, Ji et al. [2020]]).

- The above implies that one can write the function $(x, y) \mapsto \varphi(x + iy)$, where $\varphi$ is the Hermite activation, as the expectation of ReLUs such that the variance at points close to the origin is not too large.

These steps will be carried out in the supplementary material.

**Broader impact.** This work does not present any foreseeable societal consequence.

## Footnotes

*This work was partly done while R. Eldan and D. Mikulincer were visiting Microsoft Research.

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
