[Supplementary Material]

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

## A    Baum network total weight

We sketch here the calculation of the Baum network's total weight in the case that the $x_i$'s are independent uniform points on $\mathbb{S}^{d-1}$ and $y_i$ are $\pm 1$-Bernoulli distributed. We will show that the total weight is at least $n^2/\sqrt{d}$, thus more than $n$ times the optimal attainable weight given in Proposition 1.

Consider the matrix $X$ whose rows are the vectors $(x_i)_{i \leq d}$. The vector $v$ taken in the neuron corresponding to those points solves the equation $Xv = y$ and since the distribution of $X$ is absolutely continuous, we have that $X$ is invertible almost surely and therefore $v = X^{-1}y$, implying that $|v| \geq \|X\|_{OP}^{-1}\sqrt{d}$. It is well-known (and easy to show) that with overwhelming probability, $\|X\|_{\mathrm{OP}} = O(1)$, and thus $\|v\| = \Omega(\sqrt{d})$.

Observe that by normalizing the parameter $\delta$ accordingly, we can assume that $\|u\| = 1$. By definition we have $u \cdot x_i = b$ for all $i = 1, \ldots, d$. A calculation shows that with probability $\Omega(1)$ we have $b = \Theta(1/\sqrt{d})$.

Next, we claim that $|v \cdot u| \leq (1 - \rho)\|v\|$ for some $\rho = \Omega(1)$. Indeed, suppose otherwise. Denote $c = \frac{1}{d}\sum_{i \in [d]} x_i$. It is easy to check that with high probability, $\|c\| = O\left(\frac{1}{\sqrt{d}}\right)$. Note that $v \cdot c = \frac{1}{d}\sum_{i \in [d]} y_i = O(1/\sqrt{d})$. This implies that

$$b(|v \cdot u| - O(1)) \leq |v \cdot (bu - c)| \leq \sqrt{\|v\|^2 - (v \cdot u)^2}\|bu - c\| \leq \sqrt{2\rho}\frac{\|v\|}{\sqrt{d}},$$

where we used the fact that $(bu - c) \perp (v \cdot u)u$. Thus we have

$$\Omega(1 - 2\rho) = b(1 - 2\rho)\|v\| = O(\sqrt{\rho}).$$

leading to a contradiction. To summarize, we have $\|v\| = \Omega(\sqrt{d})$, $\|u\| = 1$, $|u \cdot v| \leq (1 - \rho)\|v\|$, $\rho = \Omega(1)$, and $b = O(1/\sqrt{d})$. Since spherical marginals are approximately Gaussian, if $x$ is uniform in $\mathbb{S}^{d-1}$ we have that the joint distribution of $(x \cdot u, x \cdot v)$ conditional on $v$ and $u$ is approximately $\mathcal{N}\left(0, \frac{1}{d}\begin{pmatrix} 1 & (1-\rho)\beta \\ (1-\rho)\beta & \beta \end{pmatrix}\right)$ with $\rho = \Omega(1)$ and $\beta = \Theta(d)$. Therefore, with probability $\Omega(1/n)$ we have $|x \cdot v| = \Omega(1)$ and $|x \cdot u - b| = O(1/(n\sqrt{d}))$.

We conclude that

$$\mathbb{P}\left(\exists i \geq d + 1 \text{ s.t. } \frac{|x_i \cdot u - b|}{|x_i \cdot v|} = O\left(\frac{1}{n\sqrt{d}}\right) \middle| x_1, ..., x_d\right) = \Omega(1).$$

Therefore, we get $\delta = O(1/n\sqrt{d})$ which implies that the weight of the neuron is of order at least $\frac{\|u\|}{\delta} = \Omega(n\sqrt{d})$. This happens with probability $\Omega(1)$ for every one of the first $n/(2d)$ neurons, implying that the total weight is of order $n^2/\sqrt{d}$.

## B    NTK network total weight

We sketch here the calculation of the NTK network's total weight. Recall that the neurons are of the form (9). According to Lemma 3, we have that for typical neurons, $\|v\| = \Omega(\sqrt{n})$. Moreover, with high probability we have $\|u\| = \Theta(\sqrt{d})$, and thus the weight of a single neuron is at least $\frac{\|u\|}{\delta} = \frac{\sqrt{d}}{\delta}$. Adding up the neurons, this shows that the total weight is of order $\frac{\sqrt{d}}{\delta}$ (since $k = \widetilde{\Theta}(n/d)$ and the coefficient in front of the neurons is of order $\widetilde{\Theta}(\frac{d}{n})$).

Now suppose that $\delta$ is taken according to (4). The main observation (we omit the details of proof) is that $u$ and $v$ have a mutual distribution of roughly independent Gaussian random vectors (without loss of generality we can assume that $\sum y_i = 0$ which implies $\mathbb{E}u \cdot v = 0$). In this case we have $\delta = \widetilde{O}\left(\frac{\sqrt{d}}{n\sqrt{n}}\right)$. This implies a total weight of order at least $n\sqrt{n}$.

## C  Hermite polynomials

Define the $m$'th Hermite polynomial by:

$$H_m(x) = \frac{(-1)^m}{\sqrt{m!}} \left( \frac{d^m}{dx^m} e^{-\frac{x^2}{2}} \right) e^{\frac{x^2}{2}}.$$

For ease of notion we also define $H_{-1} \equiv 0$. The Hermite polynomials may also be regarded as the power series associated to the function $F(t, x) = \exp(tx - \frac{t^2}{2})$. Indeed,

$$\begin{aligned}
F(t, x) &= \exp\left( \frac{x^2}{2} - \frac{(x-t)^2}{2} \right) \\
&= e^{\frac{x^2}{2}} \sum_{\ell=0}^{\infty} \frac{t^m}{m!} \left( \frac{d^m}{dt^m} e^{-\frac{(x-t)^2}{2}} \right) \Big|_{t=0} \\
&= \sum_{m=0}^{\infty} \frac{t^m}{\sqrt{m!}} H_m(x).
\end{aligned} \tag{13}$$

Observe that $\frac{d}{dx} F(t, x) = t F(t, x)$, so that, since $H_0 \equiv 1$,

$$\sum_{m=1}^{\infty} \frac{t^m}{\sqrt{(m-1)!}} H_{m-1}(x) = \sum_{m=1}^{\infty} \frac{t^m}{\sqrt{m!}} H'_m(x),$$

and we deduce

$$H'_m = \sqrt{m} H_{m-1}. \tag{14}$$

Also $\frac{d}{dt} F(t, x) = (x - t) F(t, x)$ and a similar argument shows that

$$\sqrt{\frac{m}{m-1}} H_m(x) = \frac{x}{\sqrt{m-1}} H_{m-1}(x) - H_{m-2}(x). \tag{15}$$

Furthermore, we show that the family $\{H_m\}$ satisfies the following orthogonality relation, which we shall freely use.

**Lemma 4** *Let $X, Y \sim \mathcal{N}(0, 1)$ be jointly Gaussian with $\mathbb{E}[XY] = \rho$. Then*
$$\mathbb{E}\left[ H_m(X) H_{m'}(Y) \right] = \delta_{m,m'} \rho^m.$$

**Proof.** Fix $s, t \in \mathbb{R}$. We have the following identity

$$\mathbb{E}\left[ F(s, X) F(t, Y) \right] = \mathbb{E}\left[ \exp(sX + tY) \right] \exp\left( -\frac{s^2 + t^2}{2} \right) = e^{st \cdot \rho},$$

where in the second equality we have used the formula for the moment generating functions of bi-variate Gaussians. In particular, we have

$$\frac{d^{m+m'}}{ds^m dt^{m'}} \mathbb{E}\left[ F(s, X) F(t, Y) \right] \Big|_{t=0, s=0} = \frac{d^{m+m'}}{ds^m dt^{m'}} e^{st \cdot \rho} \Big|_{t=0, s=0}.$$

By (13), the left hand side equals $\mathbb{E}\left[ H_\ell(X) H_{\ell'}(Y) \right]$, while the right hand side is $\delta_{m,m'} \rho^m$. The proof is complete. □

## D  More general non-linearities

We now consider an arbitrary $L$-Lipschitz non-linearity $\psi$ that is differentiable except at a finite number of points and such that $\mathbb{E}_{X \sim \mathcal{N}(0,1)}[(\psi'(X))^2] < +\infty$. In particular, with $H_1, H_2, \ldots$ being the Hermite polynomials (normalized such that it forms an orthonormal basis) we have that there exists a sequence of real numbers $(a_\ell)$ such that

$$\psi' = \sum_{\ell \geq 0} a_\ell H_\ell.$$

Our generalization of Theorem 2 now reads as follows:

**Theorem 4** *Under the above assumptions on $\psi$, there exists $f \in \mathcal{F}_k(\psi)$ with $\|\mathbf{f} - \mathbf{y}\|^2 \leq \varepsilon \|\mathbf{y}\|^2$ provided that*

$$k \cdot d \geq \frac{16\omega \cdot L}{\sum_{\ell \geq \frac{\log(2n)}{2\log(1/\gamma)}} a_\ell^2} \cdot n \log(1/\varepsilon) \,.$$

*In fact there is an efficient procedure that produces a random $f \in \mathcal{F}_k(\psi)$ with $\mathbb{E}[\|\mathbf{f} - \mathbf{y}\|^2] \leq \varepsilon \|\mathbf{y}\|^2$ when* (5) *holds.*

**Proof.** First we follow the proof of Lemma 2, with the only change being: (i) in (9) there is an additive $O(\delta)$ term (also now the condition on $u$ is that $u \cdot x_i$ is not in the finite set of points where $\psi$ is not differentiable), and (ii) in (10) we use that $|\psi'| \leq L$. We obtain that for $u \in \mathbb{R}^d$ there exists $f \in \mathcal{F}_2(\psi)$ such that

$$\sum_{i=1}^n y_i f(x_i) \geq \frac{1}{2} \left\| \sum_{i=1}^n \psi'(u \cdot x_i) y_i x_i \right\|^2 \,, \tag{16}$$

and furthermore

$$\sum_{i=1}^n f(x_i)^2 \leq \frac{2\omega \cdot n \cdot L}{d} \cdot \sum_{i=1}^n y_i f(x_i) \,, \tag{17}$$

where the added term $L$ is due to modification (ii) above and the added 2 is due to (i).

Next we follow the proof of Lemma 3, noting that the matrix $H$ is now defined by (recall Lemma 4) $H_{i,j} = \sum_{\ell \geq 0} a_\ell^2 (x_i \cdot x_j)^{\ell+1}$, to obtain:

$$\mathbb{E}_{u \sim \mathcal{N}(0, \mathrm{I}_n)} \left\| \sum_{i=1}^n \psi'(u \cdot x_i) y_i x_i \right\|^2 \geq \frac{1}{2} \sum_{\ell \geq \frac{\log(2n)}{2\log(1/\gamma)}} a_\ell^2 \cdot \sum_{i=1}^n y_i^2 \,. \tag{18}$$

In particular we obtain from (16) and (18) that (6) holds true with the term $\frac{1}{10} \cdot \sqrt{\frac{\log(1/\gamma)}{\log(2n)}}$ replaced by $\frac{1}{4} \sum_{\ell \geq \frac{\log(2n)}{2\log(1/\gamma)}} a_\ell^2$, and from (17) that (7) holds true with $\omega$ replaced by $2\omega \cdot L$. We can thus conclude as we concluded Theorem 2 from Theorem 3. $\square$

## E  More details on the complex network

We will work under the assumptions that

$$\|x_i\| = 1 \text{ for every } i \in [n], \text{ and, } |x_i \cdot x_j| \leq \gamma \text{ for } i \neq j. \tag{19}$$

In light of Lemma 1, it is enough to find a single neuron whose scalar product with the data set is large. Thus, the rest of this section is devoted to proving the following theorem.

**Theorem 5** *Assume that* (19) *holds, that $m$ is large enough so that $n\gamma^{m-2} \leq \frac{1}{2}$ and that for all $i \in [n]$, $y_i^2 \leq n\gamma^2$ with $\|\mathbf{y}\|^2 \leq n$. Then, there exist $w \in \mathbb{R}^d$ and $b, \sigma \in \mathbb{R}$, with*

$$\|w\|^2, |b|^2 \leq C_m d \log(n)^m, |\sigma| = 1,$$

*such that for*

$$f(x) = \sigma \cdot \mathrm{ReLU}\big(w \cdot x + b\big),$$

*we have*

$$\mathbf{y} \cdot \mathbf{f} \geq \frac{c_m}{\log(n)^{m^2/2}} \frac{1}{\sqrt{n\gamma^2}} \|\mathbf{y}\|^2,$$

*and*

$$\|\mathbf{f}\|^2 \leq \frac{n}{c_m} \log(n)^m,$$

*where $c_m, C_m > 0$ are constants which depends only on $m$.*

Equipped with the above result one can prove the following formal version of Theorem 1. If $A \subset [n]$ and $v \in \mathbb{R}^n$ we denote below by $v_A$ the projection of $v$ unto the indices contained in $A$. With this notation our result is:

**Theorem 6** *Assume that* (19) *holds, that $m$ is large enough so that $n\gamma^{m-2} \leq \frac{1}{2}$ and that $\|\mathbf{y}\|^2 = n$. There exists $f \in \mathcal{F}_k(\mathrm{ReLU})$ and $A \subset [n]$, with*

$$k = \left\lceil C_m \gamma^2 \frac{\log(1/\varepsilon)}{\varepsilon} n \log(n)^{(m^2+m)} \right\rceil,$$

*such that*

$$\mathbb{E}[\|\mathbf{f}_A - \mathbf{y}_A\|^2] \leq \varepsilon \|\mathbf{y}\|^2, \quad |A| \geq n - \frac{1}{\gamma^2}, \tag{20}$$

*and*

$$\mathbf{W}(f) = \widetilde{O}\left(\frac{\log(1/\varepsilon)}{\varepsilon}\sqrt{n\gamma^2 d}\right), \tag{21}$$

*where $C_m$ is a constant which depends only on $m$.*

Observe that if $(x_i)_{i \in [n]}$ are uniformly distributed in the $\mathbb{S}^{d-1}$ then $\gamma = \widetilde{O}\left(\frac{1}{\sqrt{d}}\right)$ and we get that $\mathbf{W}(f) = \widetilde{O}\left(\frac{\log(1/\varepsilon)}{\varepsilon}\sqrt{n}\right)$, which is optimal up to the logarithmic factors and the dependence on $\varepsilon$.

The proof of Theorem 6 follows an iterative procedure similar to the one carried out in Lemma 1. The only caveat is the condition $y_i^2 \leq n\gamma^2$ which appears in Theorem 5. Due to this condition we need to consider a slightly smaller set of indices at each iteration, ignoring ones where the residue becomes too big.

**Proof.**[of Theorem 6] We build the network iteratively. Set $f_0 \equiv 0$, $A_0 = [n]$ and $r_{0,i} = y_i$. Now, for $\ell \in \mathbb{N}$, suppose that there exists $f_\ell \in \mathcal{F}_\ell(\mathrm{ReLU})$ with

$$\|(\mathbf{f}_\ell)_{A_\ell} - \mathbf{y}_{A_\ell}\| \leq \left(1 - \frac{c_m^3}{\log(n)^{m^2+m}} \frac{\varepsilon}{n\gamma^2}\right)\|\mathbf{y}\|^2.$$

Set $r_{\ell,i} = y_i - f_\ell(x_i)$ and $A_\ell = \{i \in A_{\ell-1} | r_{\ell,i}^2 \leq n\gamma^2\}$. We now invoke Theorem 5 with the residuals $\{r_{\ell,i} | i \in A_\ell\}$ to obtain a neuron $f \in \mathcal{F}_1(\mathrm{ReLU})$, which satisfies

$$(\mathbf{r}_\ell)_{A_\ell} \cdot \mathbf{f} \geq \frac{c_m}{\log(n)^{m^2/2}} \frac{1}{\sqrt{n\gamma^2}}\|\mathbf{r}_\ell\|^2,$$

and

$$\|\mathbf{f}_{A_\ell}\|^2 \leq \frac{n}{c_m}\log(n)^m.$$

Since we may assume $\|(\mathbf{r}_\ell)_{A_\ell}\|^2 \geq n\varepsilon$ (otherwise we are done), the second condition can be rewritten as

$$\|\mathbf{f}_{A_\ell}\|^2 \leq \frac{\log(n)^m}{c_m \varepsilon}\|(\mathbf{r}_\ell)_{A_\ell}\|^2.$$

In this case the calculation done in Lemma 1 with $\alpha = \frac{c_m}{\log(n)^{m^2/2}}\frac{1}{\sqrt{n\gamma^2}}$ and $\beta = \frac{\log(n)^m}{c_m \varepsilon}$ shows that for $\eta := \frac{c_m^2 \varepsilon}{\log(n)^{m^2/2+m}}$, one has

$$\|\eta \mathbf{f}_{A_\ell} - (\mathbf{r}_\ell)_{A_\ell}\|^2 \leq \left(1 - \frac{c_m^3}{\log(n)^{m^2+m}}\frac{\varepsilon}{n\gamma^2}\right)\|(\mathbf{r}_\ell)_{A_\ell}\|^2.$$

In other words, if we define $f_{\ell+1} \in \mathcal{F}_{\ell+1}(\mathrm{ReLU})$ by $f_{\ell+1} = f_\ell + \eta f$,

$$\|(\mathbf{f}_{\ell+1})_{A_\ell} - \mathbf{y}_{A_\ell}\|^2 \leq \left(1 - \frac{c_m^3}{\log(n)^{m^2+m}}\frac{\varepsilon}{n\gamma^2}\right)^{\ell+1}\|\mathbf{y}\|^2.$$

The estimate (20) is now obtained with the appropriate choice of $k$. Let us also remark that for any $\ell$,

$$\|(\mathbf{r}_{\ell+1})_{A_\ell}\|^2 \leq \|(\mathbf{r}_\ell)_{A_\ell}\|^2 \leq \|(\mathbf{r}_\ell)_{A_{\ell-1}}\|^2 - n\gamma^2|A_{\ell-1} \setminus A_\ell|.$$

By induction

$$\|(\mathbf{r}_{\ell+1})_{A_\ell}\|^2 \leq \|\mathbf{y}\|^2 - n\gamma^2 (n - |A_\ell|)$$

This shows that $|A_\ell| \geq n - \frac{1}{\gamma^2}$. The bound on $\mathbf{W}(f_k)$ a direct consequence of Lemma 1. $\qquad\square$

### E.1 Constructing the complex neuron

In the sequel, we fix $m \in \mathbb{N}$ so that

$$n\gamma^{m-2} \leq \frac{1}{2}. \tag{22}$$

Define

$$\varphi(z) = \frac{1}{\sqrt{m}} H_m(z), \;\; z \in \mathbb{C}$$

where $H_m$ is the $m$-th Hermite polnomial. Note that we also have $\varphi' = H_{m-1}$.

We first prove the following analogous result to Theorem 5 where $\psi$ is replaced by $\varphi$.

**Lemma 5** *Assume that* (19), (22) *hold, and that for all $i \in [n]$, one has $y_i^2 \leq n\gamma^2$. There exist $\widetilde{w}, \widetilde{w}' \in \mathbb{R}^d$ and $z \in \mathbb{C}, |z| = 1$, such that for*

$$g(x) = Re\left(z \cdot \varphi\left(\left(\widetilde{w} + \mathbf{i}\widetilde{w}'\right) \cdot x\right)\right), \tag{23}$$

*we have,*

$$\mathbf{y} \cdot \mathbf{g} \geq \frac{1}{2C_m\sqrt{n\gamma^2}}\|\mathbf{y}\|^2.$$

*Moreover, its weights admit the bounds*

$$\|\widetilde{w}\|^2, \|\widetilde{w}'\|^2 \leq d(4C_m \log(n))^m \tag{24}$$

*and for all $i \in [n]$,*

$$|\widetilde{w} \cdot x_i|, |\widetilde{w}' \cdot x_i| \leq (4C_m \log(n))^{\frac{m}{2}}.$$

Our approach to Lemma 5 will be to construct an appropriate distribution on neurons of type (23), and then show that the desirable properties are attained with positive probability. In what follows, let $w \sim \mathcal{N}(0, \mathrm{I}_d)$. Define

$$v(w) := \frac{1}{\sqrt{n\gamma^2}} \sum_{i=1}^n y_i\varphi'(w \cdot x_i)x_i.$$

Next, let $a$ be uniformly distributed in the complex unit circle, and finally define

$$g(x) = \mathrm{Re}\left(a^{-1}\varphi((w + av(w)) \cdot x)\right). \tag{25}$$

We will prove the following two bounds.

**Lemma 6** *Under the assumptions* (19) *and* (22), *one has*

$$\mathbb{E}\left[\mathbf{y} \cdot \mathbf{g}\right] \geq \frac{1}{2\sqrt{n\gamma^2}}\|\mathbf{y}\|^2.$$

**Lemma 7** *Suppose that the assumptions* (19) *and* (22) *hold. Assume also that for every $i$ we have $y_i \leq n\gamma^2$. Then one has, for a constant $C_m > 0$ which depends only on $m$,*

$$\mathbb{E}[\|\mathbf{g}\|^2] \leq C_m n.$$

*Moreover, for every $i \in [n]$ and $s > s_0$, for some constant $s_0$,*

$$\mathbb{P}\left(|\mathrm{Re}((w + v(w)) \cdot x_i| > s\right), \mathbb{P}\left(|\mathrm{Im}((w + v(w))) \cdot x_i| > s\right) \leq \exp\left(\frac{1}{C_m}s^{-2/m}\right). \tag{26}$$

Recall the definition of the Gram matrix $H$,

$$H_{i,j} = \mathbb{E}_{w \sim \mathcal{N}(0, \mathrm{I}_d)} \left[ \varphi'(w \cdot x_i) \varphi'(w \cdot x_j) x_i \cdot x_j \right].$$

As suggested in (12), we will need to bound $H$ from below. We will need the following lemma.

**Lemma 8** *Under the assumptions* (19) *and* (22)*, one has* $H \succeq \frac{1}{2} \mathrm{I}_n$.

**Proof.** If $X$ and $Y$ are standard, jointly-normal random variables with $\mathbb{E}[XY] = \rho$, by Lemma 4 one has $\mathbb{E}[H_{m-1}(X) H_{m-1}(Y)] = \rho^{m-1}$ and thus here $H_{i,j} = (x_i \cdot x_j)^m$. In particular if $n \cdot \gamma^m \leq 1/2$ we obtain that for all $i \in [n]$ one has $1 = H_{i,i} \geq 2 \sum_{j \neq i} |H_{i,j}|$. By diagonal dominance we conclude that $H \succeq \frac{1}{2} \mathrm{I}_n$. $\qquad \square$

**Proof.** [Proof of Lemma 6] For any $\beta \in \mathbb{N}, \beta \neq 1$, we have that $\mathbb{E}\left[a^{-1+\beta}\right] = 0$. Thus, since $\varphi$ is an entire function, by taking its Taylor expansion around the point $w$, we obtain the identity

$$\mathbb{E}_a \left[ a^{-1} \varphi((w + av(w)) \cdot x) \right] = \sum_{\beta=0}^{\infty} \frac{1}{\beta!} \mathbb{E}_a \left[ a^{-1+\beta} \varphi^{(\beta)}(w \cdot x_i)(v(w) \cdot x)^{\beta} \right] = \varphi'(w \cdot x) v(w) \cdot x.$$

So we can estimate

$$\mathbb{E}_{w,a} \left[ \sum_{i=1}^{n} y_i \mathrm{Re} \left( a^{-1} \varphi((w + av(w)) \cdot x_i) \right) \right] = \sum_{i=1}^{n} y_i \mathbb{E}_w \left[ \varphi'(w \cdot x_i) v(w) \cdot x_i \right]$$

$$= \frac{1}{\sqrt{n\gamma^2}} \sum_{i,j} y_i y_j \mathbb{E}_w \left[ \varphi'(w \cdot x_i) \varphi'(w \cdot x_j) x_i \cdot x_j \right]$$

$$= \frac{1}{\sqrt{n\gamma^2}} \mathbf{y}^{\top} H \mathbf{y} \geq \frac{1}{2\sqrt{n\gamma^2}} \|\mathbf{y}\|^2,$$

where the last inequality follows from Lemma 8. $\qquad \square$

**Proof.** [Proof of Lemma 7] In what follows, the expression $C_m$ will denote a constant depending only on $m$, whose value may change between different appearances. Our objective is to obtain an upper bound on

$$\|\mathbf{g}\|^2 = \sum_{i=1}^{n} |\mathrm{Re} \left( a^{-1} \varphi((w + av(w)) \cdot x_i) \right)|^2.$$

Since $\varphi$ is a polynomial of degree $m$ we have

$$\|\mathbf{g}\|^2 \leq C_m \sum_{i=1}^{n} \left( 1 + |w \cdot x_i|^{2m} + |v(w) \cdot x_i|^{2m} \right).$$

Moreover $w \cdot x_i$ is a standard Gaussian and thus $\mathbb{E}[|w \cdot x_i|^{2m}] \leq (2m)^m$. It therefore remains to control, for $x \in \{x_1, \ldots, x_n\}$, the expression

$$|v(w) \cdot x|^{2m} = \frac{1}{(n\gamma^2)^m} \left| \sum_{i=1}^{n} y_i H_{m-1}(w \cdot x_i) x_i \cdot x \right|^{2m}.$$

From hypercontractivity and the fact that the Hermite polynomials are eigenfunctions of the Ornstein-Uhlenbeck operator we have (see [Janson, 1997, Theorem 5.8])

$$\mathbb{E}\left[ |v(w) \cdot x|^{2m} \right] \leq (2m)^{2m^2} \mathbb{E}\left[ |v(w) \cdot x|^2 \right]^m.$$

Thus, it will be enough to show that $\mathbb{E}_w[|v(w)\cdot x_j|^2] = O(1)$. We calculate

$$
\mathbb{E}_w[|v(w)\cdot x_j|^2] = \frac{1}{n\gamma^2}\mathbb{E}\left|\sum_{i=1}^n y_i H_{m-1}(w\cdot x_i) x_i \cdot x_j\right|^2
$$

$$
= \frac{1}{n\gamma^2}\left(\mathbb{E}\sum_{i=1}^n y_i^2 \mathbb{E}[(H_{m-1}(w\cdot x_i))^2]|x_i\cdot x_j|^2\right.
$$

$$
\left. + \sum_{i\neq i'} y_i y_{i'}\mathbb{E}[H_{m-1}(w\cdot x_i)H_{m-1}(w\cdot x_i')](x_i\cdot x_j)(x_{i'}\cdot x_j)\right)
$$

$$
\leq \frac{1}{n\gamma^2}\left(\sum_{i=1}^n y_i^2|x_i\cdot x_j|^2 + \frac{\gamma^{m-1}}{n\gamma^2}\sum_{i\neq i'}|y_i y_{i'}(x_{i'}\cdot x_j)(x_i\cdot x_j)|\right),
$$

where we used that $\mathbb{E}[(H_{m-1}(w\cdot x_i))^2] = 1$ and

$$
|\mathbb{E}[H_{m-1}(w\cdot x_i)H_{m-1}(w\cdot x_{i'})]| = |x_i\cdot x_{i'}|^{m-1} \leq \gamma^{m-1},
$$

valid whenever $i\neq i'$. By using that $\|\mathbf{y}\|^2 = O(n)$, we get

$$
\frac{1}{n\gamma^2}\sum_{i=1}^n y_i^2|x_i\cdot x_j|^2 \leq \frac{y_j^2}{n\gamma^2} + \frac{\|\mathbf{y}\|^2}{n} = O(1).
$$

To deal with the last term, observe that since $i\neq i'$ then $|(x_{i'}\cdot x_j)(x_i\cdot x_j)| \leq \gamma$, thus

$$
\frac{\gamma^{m-1}}{n\gamma^2}\sum_{i\neq i'}|y_i y_{i'}(x_{i'}\cdot x_j)(x_i\cdot x_j)| \leq \frac{\gamma^{m-2}}{n}\left(\sum_{i=1}^n |y_i|\right)^2 \leq \gamma^{m-2}\|\mathbf{y}\|^2 = O(1),
$$

where in the last inequality we've used $\gamma^{m-2} \leq \frac{1}{n}$. So, $\mathbb{E}_w[|v(w)\cdot x_i|^2] = O(1)$ as required.

Finally, to see (26) observe that both $\mathrm{Re}(w + v(w))$ and $\mathrm{Im}(w + v(w))$ are given by degree $m$ polynomials of $w$, a standard Gaussian random vector. In [Janson, 1997, Theorem 6.7] it is shown that there exists a constant $a_m$ depending only on $m$, such that if $P$ is a polynomial of degree $m$ and $X$ is a standard normal random variable, then for every $t > 2$,

$$
\mathbb{P}\left(|p(X)| > t\sqrt{\mathbb{E}[p(X)^2]}\right) \leq \exp\left(-a_m t^{2/m}\right)
$$

Thus, since

$$
\mathbb{E}\left[|\mathrm{Re}(w + v(w))\cdot x_i|^2\right], \mathbb{E}\left[|\mathrm{Im}(w + v(w))\cdot x_i|^2\right] \leq \mathbb{E}\left[1 + |w\cdot x_i|^{2m} + |v(w)\cdot x_i|^{2m}\right] \leq C_m,
$$

the bound (26) follows. $\qquad\square$

We are finally ready to prove the existence of the complex neuron.
**Proof.**[Proof of Lemma 5] Consider the random variable

$$
F = \mathbf{g}\cdot\mathbf{y} = \sum_{i=1}^n y_i g(x_i)
$$

and set $W = \mathrm{Re}(w + v(w))$ and $W' = \mathrm{Im}(w + v(w))$. Lemma 6 gives

$$
\mathbb{E}[F] \geq \frac{1}{2\sqrt{n\gamma^2}}\|\mathbf{y}\|^2.
$$

Using Lemma 7 and Cauchy-Schwartz we may see that

$$
\mathbb{E}\left[F^2\right] \leq \sum_{i=1}^n y_i^2 \mathbb{E}_{w,a}\left[\sum_{i=1}^n g(x_i)^2\right] \leq C_m n\|\mathbf{y}\|^2.
$$

Define $G = \mathbb{1}_{\left\{ \exists i : |W \cdot x_i|, |W' \cdot x_i| \geq (4C_m \log(n))^{\frac{m}{2}} \right\}}$. A second application of Cauchy-Schwartz gives

$$\mathbb{E}\Big[FG\Big] \leq \sqrt{C_m n \|\mathbf{y}\|^2 \mathbb{E}\left[G\right]}.$$

Now, the estimate (26) and a union bound yields

$$\mathbb{E}\left[G\right] \leq n \exp\left(-4\log(n)\right) \leq \frac{1}{n^3}.$$

Therefore,

$$\mathbb{E}\Big[FG\Big] \leq \frac{1}{n} C_m \|\mathbf{y}\|.$$

Combining this with the lower bound of $\mathbb{E}[F]$, we finally have

$$\mathbb{E}\Big[F(1-G)\Big] \geq \frac{1}{2\sqrt{n\gamma^2}} \|\mathbf{y}\|^2 - \frac{1}{n} C_m \|\mathbf{y}\| \geq \frac{1}{4\sqrt{n\gamma^2}} \|\mathbf{y}\|^2,$$

where the last inequality is valid as long as $n$ is large enough. The claim now follows via taking a realization that exceeds the expectation. Since we might as well assume that the sample contains an orthonormal basis, (24) follows as well. $\qquad\square$

## E.2 Approximating a complex neuron with ReLU activation

Our goal in this section is to prove the following lemma, showing that the complex polynomial can be essentially replaced by a ReLU. We write $\psi(t) = \mathrm{ReLU}(t)$ and recall that $\varphi(t) = \frac{1}{\sqrt{m}} H_m(t)$.

**Lemma 9** *For any $w, w' \in \mathbb{R}^d, z \in \mathbb{C}$ with $|z| = 1$ and $M > 0$, there exist a pair of random variables $S, B$ and a random vector $W \in \mathbb{R}^d$ such that for any $x \in \mathbb{S}^{d-1}$ with $m\left(|w \cdot x| + |w' \cdot x|\right) \leq M$,*

$$\mathbb{E}\left[S\psi(W \cdot x - B)\right] = \frac{c_{z,m}}{M^m} \mathrm{Re}\left(z \cdot \varphi(w \cdot x + \mathbf{i}w' \cdot x)\right),$$

*where $c_{z,m}$ depends only on $m$ and $z$ and there exists another constant $c_m$, such that*

$$\frac{1}{c_m} \geq c_{z,m} \geq c_m. \tag{27}$$

*Moreover,*

$$|S| = 1, |B| \leq M \quad \text{almost surely,}$$

*and*

$$W = w + j \cdot w' \text{ for some } j \in \{0, 1, \ldots, m\}.$$

Let us first see how to complete the proof of Theorem 5 using the combination of the above with Lemma 5.

**Proof.**[Proof of Theorem 5] Invoke Lemma 5 to obtain a function

$$g(x) = Re\left(z \cdot \varphi\left(x \cdot \widetilde{w} + \mathbf{i}x \cdot \widetilde{w}'\right)\right)$$

such that

$$\mathbf{y} \cdot \mathbf{g} \geq \frac{1}{2C_m\sqrt{n\gamma^2}} \|\mathbf{y}\|^2,$$

and such that for every $i \in [n]$,

$$|\widetilde{w} \cdot x_i|, |\widetilde{w}' \cdot x_i| \leq C_m \log(n)^{\frac{m}{2}}.$$

Set $M = 2C_m m \log(n)^{\frac{m}{2}}$, so that $m(|\widetilde{w} \cdot x_i| + |\widetilde{w}' \cdot x_i|) \leq M$. By Lemma 9, we may find $\sigma, w, b$, such that

$$|b|^2 \leq M^2, \|w\|^2 \leq m^2(\|\widetilde{w}\| + \|\widetilde{w}'\|)^2 \leq 4C_m m^2 d \log(n)^m, \quad |\sigma| = 1,$$

for which we define $f(x) = \sigma\psi(w \cdot x - b)$. The lemma then implies,

$$\mathbf{y} \cdot \mathbf{f} \geq \frac{c_m}{M^m}\mathbf{y} \cdot \mathbf{g} \geq \frac{c'_m}{M^m\sqrt{n\gamma^2}}\|\mathbf{y}\|^2,$$

and

$$\|\mathbf{f}\|^2 = \sum_{i=1}^{n}(\psi(w \cdot x_i - b))^2 \leq 2\sum_{i=1}^{n}\left(|w \cdot x_i|^2 + b^2\right)$$

$$\leq 2M^2n + 2\sum_{i=1}^{n}|w \cdot x_i|^2.$$

By Lemma 9, $w = \widetilde{w} + j \cdot \widetilde{w}'$ for some $j = 0, ..., m$. Hence, $|w \cdot x_i|^2 \leq 2m(|\widetilde{w} \cdot x_i|^2 + |\widetilde{w}' \cdot x_i|^2)$ and

$$\|\mathbf{f}\|^2 \leq 2M^2n + 4m\sum_{i=1}^{n}(|\widetilde{w} \cdot x_i|^2 + |\widetilde{w} \cdot x_i|^2) \leq 10mM^2n.$$

The proof is concluded by substituting $M$. $\qquad\square$

It remains to prove Lemma 9. This is done in the next subsections.

### E.2.1 On homogeneous polynomials

Since our aim is to approximate a polynomial by ReLU, we first find an appropriate polynomial basis to work with.

**Lemma 10** *Any polynomial of the form $(x, y) \to Re(z \cdot (x + \mathbf{i}y)^m)$ has the form,*

$$\sum_{j=0}^{m} a_j(x + j \cdot y)^m.$$

**Proof.** Define

$$\mathcal{H}_m = \{p(x, y)|p \text{ is a degree } m \text{ homogeneous polynomial}\},$$

and

$$A_m = \{(x + j \cdot y)^m|j = 0, ..., m\}.$$

It will suffice to show that $A_m$ forms a basis for $\mathcal{H}_m$. The result will follow since $Re(z \cdot (x + \mathbf{i}y)^m)$ is clearly homogeneous. For $0 \leq j \leq m$, set $p_j = (x + j \cdot y)^m$, so that $p_j \in A_m$ and

$$p_j = \sum_{k=0}^{m}\binom{m}{k}j^k y^k x^{m-k}.$$

Note that the set $\{\binom{m}{k}y^k x^{m-k}|k = 0, ..., m\}$ forms a basis for $\mathcal{H}_m$ and in that basis $p_j$ has coordinates $(1, j, ..., j^m)$. Taking the Vandermonde determinant of the matrix whose columns are $\{p_j : j = 0, ..., m\}$, we see that it must also be a basis for $\mathcal{H}_m$. $\qquad\square$

**Corollary 1** *Let $w, w' \in \mathbb{R}^d$ and $z \in \mathbb{C}$, then we have the following representation:*

$$\mathrm{Re}\left(z \cdot \varphi(w \cdot x + \mathbf{i}w' \cdot x)\right) = \sum_{j=0}^{m} p_{z,j}((w + jw') \cdot x),$$

*where each $p_{z,j}$ is a polynomial of degree $m$, which depends continuously on $z$.*

**Proof.** The representation is immediate from the previous lemma. To address the point of continuity, we write

$$\mathrm{Re}\left(z \cdot \varphi(w \cdot x + \mathbf{i}w' \cdot x)\right) = \mathrm{Re}(z)\mathrm{Re}\left(\varphi(w \cdot x + \mathbf{i}w' \cdot x)\right) - \mathrm{Im}(z)\mathrm{Im}\left(\varphi(w \cdot x + \mathbf{i}w' \cdot x)\right)$$

$$= \mathrm{Re}(z)\mathrm{Re}\left(\varphi(w \cdot x + \mathbf{i}w' \cdot x)\right) + \mathrm{Im}(z)\mathrm{Re}\left(\mathbf{i} \cdot \varphi(w \cdot x + \mathbf{i}w' \cdot x)\right)$$

$$= \sum_{j=0}^{m}\left(\mathrm{Re}(z)p_{1,j}((w + jw') \cdot x) + \mathrm{Im}(z)p_{\mathbf{i},j}((w + jw') \cdot x)\right).$$

So, $p_{z,j}$ is a linear combination of $p_{1,j}$ and $p_{\mathbf{i},j}$, with coefficients that vary continuously in $z$. $\qquad\square$

### E.2.2 ReLUs as universal approximators

Next, we show how ReLU functions might be used to universally approximate compactly supported functions.

**Proposition 5** *Let $f : \mathbb{R} \to \mathbb{R}$ be twice differentiable and compactly supported on $[-M, M]$. Then, there exists a pair of random variables $S, B$, such that, for every $x \in [-M, M]$,*

$$\mathbb{E}\left[S\psi(x - B)\right] = \frac{f(x)}{\int |f''|},$$

*and such that, almost surely $|B| \leq M$ and $|S| = 1$.*

**Proof.** Observe that, when considered as a distribution, $\psi(x)'' = \delta_0$. Therefore, there exists a linear function $L$ such that

$$f(x) + L(x) = \int_{-M}^{M} \psi(x - y)f''(y)dy.$$

$f''(x)$ is the second derivative of a compactly supported function which implies that $f(x) + L(x)$ is compactly supported as well. Hence, $L(x) \equiv 0$. Let $B$ be the random variable whose density is $\frac{|f''|}{\int_{-M}^{M}|f''|}$ and set $S = \text{sign}(f''(B)))$. We now have

$$\mathbb{E}\left[S\psi(x - B)\right] = \frac{\int_{-M}^{M} \psi(x - y)f''(y)dy}{\int_{-M}^{M}|f''|} = \frac{f(x)}{\int |f''|}.$$

$\square$

### E.2.3 Completing the proof of Lemma 9

Set $\chi_M$ to be a bump function for the interval $[-M, M]$. That is,

- $\chi_M : \mathbb{R} \to \mathbb{R}$ is smooth.
- $0 \leq \chi_M \leq 1$.
- $\chi_M(x) = 1$ for $x \in [-M, M]$.
- $\chi_M(x) = 0$ for $|x| > 2M$.

By Corollary 1, for any $w, w' \in \mathbb{R}^d, z \in \mathbb{C}$ we have the representation

$$\text{Re}\left(z \cdot \varphi(w \cdot x + \mathrm{i}w' \cdot x)\right)\chi_M(|w \cdot x| + m|w' \cdot x|) = \sum_{j=0}^{m} p_{z,j}((w + jw') \cdot x)\chi_M(|w \cdot x| + m|w' \cdot x|).$$

$$(28)$$

**Proof.**[Proof of Lemma 9] Define $X = \left\{x \in \mathbb{S}^{d-1}; m\left(|w \cdot x| + |w' \cdot x|\right) \leq M\right\}$. Observe that for all $x \in X$,

$$\text{Re}\left(\varphi(w \cdot x + \mathbf{i}w' \cdot x)\right) = \text{Re}\left(\varphi(w \cdot x + \mathbf{i}w' \cdot x)\right)\chi_M\left(m\left(|w \cdot x| + |w' \cdot x|\right)\right).$$

Moreover, if $j = 0, \ldots, m$, then $\chi_M((w + jw') \cdot x) = 1$, as well. By invoking Proposition 5 we deduce that for every $j = 0, ..., m$, there exists a pair of random variables $S_j, B_j$ and a constant $c_{z,j} > 0$ depending only on $j, m$ and $z$, such that

$$\mathbb{E}\left[S_j\psi\left((w + jw') \cdot x - B_j\right)\right] = \frac{c_{z,j}}{M^m}p_{z,j}((w + jw') \cdot x)\chi_M((w + jw') \cdot x), \ \ \forall x \in X,$$

Here we have used the fact that if $p_j$ is one of the degree $m$ polynomials in the decomposition (28), then there exist some constants $C'_{z,j}, C_{z,j} > 0$, for which

$$C'_{z,j}M^m \leq \int\limits_{-M}^{M} |p''_{z,j}| \leq \int\limits_{-2M}^{2M} |p''_{z,j}| \leq C_{z,j}M^m.$$

We now set $J$ to be a random index from the set $\{0, \ldots, m\}$ with

$$\mathbb{P}(J = j) = \frac{c_{z,j}^{-1}}{\sum\limits_{j'} c_{z,j'}^{-1}}.$$

If we set $c_{z,m} = \frac{1}{\sum\limits_{j'} c_{z,j'}^{-1}}$, and $S := S_J, B = B_J, W = w + Jw'$ it follows from (28) that

$$\mathbb{E}\left[S\psi(W \cdot x - B)\right] = \frac{c_{z,m}}{M^m} \sum_{j=0}^{m} p_{z,j}((w + jw') \cdot x)\chi_M((w + jw') \cdot x)$$

$$= \frac{c_{z,m}}{M^m} \operatorname{Re}\left(z \cdot \varphi(w \cdot x + \mathbf{i}w' \cdot x)\right) \chi_M\left(m\left(|w \cdot x| + |w' \cdot x|\right)\right).$$

Finally since, by Corollary 1, $c_{z,m}$ depends continuously on $z$, a compactness argument implies (27). $\qquad\square$