[Reviews · NeurIPS 2020]

Review 1

Summary and Contributions: Given an arbitrary dataset of N points, the paper shows existence (via construction) of two layer NN architectures that can memorize the dataset. The main contribution is threefold: 1. The authors were able to memorize using an almost optimal (upto log factors and multiplicative constants) number of Relu neurons arranged in a two layer network. 2. They were able to extend to real valued outputs instead of binary outputs dealt in the prior work. 3. They also provide a construction (using complex numbers) that can both memorize the dataset and have small weights. ====== Post rebuttal and discussion ===== I am happy with authors' answers to my questions. I am increasing my score appropriately!

Strengths: The main strength of the paper is technical ideas and proof techniques. I think that the idea of using finite differencing to express gradient and correspondingly hyperplanes to embed real valued labels was quite interesting. Similarly, using complex numbers to cancel off higher order moments is also fascinating. I enjoyed reading the proofs, and I think they can be of general interest in design of NN architectures and non-linear activations.

Weaknesses: In the following, I am clubbing together both weaknesses and my questions here. I would appreciate some feedback on them : 1. It is not clear whether SGD or GD will be able to find the optimal set of weights for the corresponding architectures developed in the paper. I think it is important that the corresponding construction should be learnable using gradient based (to avoid dependent on d) methods. Do you think it is the case here? 2. The authors use the work Boosting at multiple places (abstract, in the discussion before Lemma 1, etc). The boosting part is not clear to me? What is the underlying game? 3, Notion of error: This work considers multiplicative error model to define memorization (for eg in Lemma 1). Why? It seems that the gap between the multiplicative and the additive model can be bad by a factor of n, i.e., \eps_{additive} = \eps_{\multiplicative} \times n. Would the provided constructions be optimal in the additive model of memorization error as well ? 4. The proof of Lemma 1 does not seem to be complete (and there is no full proof in the appendix). I was not able to see it through. Can you please elaborate? 5. Proof of proposition 1 was very cute!! 6. Please elaborate more on the similarities / differences between the current work and Ji and Telgarsky 2020. 7. In Proposition 2, it is not clear why the activation should be non-polynomial. Please elaborate. 8. The authors do not provide a clear definition of the NTK regime anywhere in the paper or the appendix. I had to skip this section while reviewing because I could not understand what the term means. 9. What is the role of the parameter m in Theorem 4? 10. Is the bound on W(f) in eqn 21 in Theorem 6 optimal?

Correctness: The proof and technical ideas seem to be correct.

Clarity: The paper was clear (except for Section 4 where the background was not well established). Please take a look at my comments in the weakness section for more details.

Relation to Prior Work: Discussed well. Would appreciate more comments comparing this work to Ji and Telgarsky 2020.

Reproducibility: Yes

Additional Feedback: ====== Post rebuttal and


Review 2

Summary and Contributions: This paper considers memorization of data using neural networks, and aims to achieve the smallest total weight rather than just smallest number of neurons. Indeed, the paper (approximately) memorizes n well-separated data points with order \sqrt{n} weight, and shows that this is optimal.

Strengths: The question considered in the paper is interesting/relevant and the results are both elegant and sharp in most parameters of interest. The main technical ideas are to incrementally build a network one neuron at a time in order to iteratively reduce the approximation error, and the complex weights which result in high order Taylor expansion terms disappearing in expectation (these terms arise in analysis of correlation between a neuron and the data). The fact that the expectation for complex perturbation of the weights precisely is zeroed out is quite nice, yet as pointed out in the paper is not possible for ReLU due to it not having a holomorphic extension. This difficulty is overcome through a sequence of representation results, where polynomial activations are used but then expressed in terms of ReLUs. I am confident that this paper will be of interest to the NeurIPS community.

Weaknesses: No major weaknesses, aside from (I suppose) trying to determine optimal dependence on epsilon in addition to n.

Correctness: All of the proofs appear to be correct.

Clarity: The paper is exceptionally well-written.

Relation to Prior Work: There is a thorough comparison to prior work, except as follows: Conceptually, there are two main ideas in the paper: 1. complex weights and 2. iterative construction. I saw a somewhat related COLT 2020 paper by Bresler and Nagaraj https://arxiv.org/abs/2002.00274, which also used complex weights (via the discrete Fourier transform) and whose main idea as far as I could tell was a similar iterative construction. If we try to unpack their Theorem 1 pg 7 to see what is the total weight of their construction, it seems they have |F(\xi)| = O(sqrt{n}) and \|\xi\| = O(sqrt{d}) so O(sqrt{nd}) weight. Their result is worse in dependence on dimension but allows for more general data points, and assuming what I wrote is correct it would be appropriate to mention this and give a comparison. It seems important to explain precisely the relationship between the ideas in that paper as compared to yours, especially surrounding the iterative construction of networks. It would be good to explain also how the NTK ideas relate to those of the paper by Daniely https://arxiv.org/abs/1911.09873.

Reproducibility: Yes

Additional Feedback:


Review 3

Summary and Contributions: This work studies how many neurons and how large the total weights of a neural net is required for memorizing training data. One of the results is about constructing a ReLU neural network with $4n/d$ number of neurons for exactly memorizing the data, where $n$ is number of samples and $d$ is the dimension. Another result is showing that a ReLU neural network in the NTK regime can approximately memorize the data with O(n/d \log(1/\epsilon)) neurons, where \epsilon is the approximation error. On the other hand, the authors point out that the total weight of neural networks for these memorization results could be very large. Yet, existing theory results suggest that networks with small total weights might be able to generalize better. So the authors consider constructing a neural net with (small) optimal total weights for memorizing data.

Strengths: The paper consider several settings of memorizing training data. The number of neurons and the size of total weights are basically optimal. The claims seem to be sound and the analysis seems to be non-trivial.

Weaknesses: I think the authors should compare this paper with [Bresler and Nagaraj 2020] in details (e.g. contributions, approach of network constructions), because I feel that the techniques used in both papers are quite similar. For example, the technique that iteratively constructs sub-networks to reduce the residual errors (Lemma 1 in this paper) also appears in [Bresler and Nagaraj 2020]. Furthermore, the complex trick in Section 5 of this paper also appears in [Bresler and Nagaraj 2020]. Specifically, it seems that Lemma 9 of this paper (i.e. replacing a complex neuron with ReLU) has a similar counterpart in [Bresler and Nagaraj 2020] (Lemma 4: replacing sinusoids by ReLU). I also have a concern about the writing (see below). But I am happy to increase the score if the authors respond well. Ref: Guy Bresler and Dheeraj Nagaraj. "A Corrective View of Neural Networks: Representation, Memorization and Learning" COLT 2020.

Correctness: seems to be correct

Clarity: The writing seems to be a little bit sloppy. Here are some examples which I hope that the authors can address. Line 415 in supplementary: "...where we used the fact that (\alpha u - c ) \perp u ..." But \alpha seems not defined. Line 540-541 in supplementary: "...the expression C_m will denote a constant depending only on m, whose value may change between different appearances..." I see that C_m is used in the inequalities on line 550 and line 551. But it is unclear if it really only depends on $m$. Does C_m depend on $n$ or $\gamma$? I suggest that the authors to write down what C_m really is. Minor: line 175: Is it 2[n / 2d] or 2[n/d]?

Relation to Prior Work: See *weakness*.

Reproducibility: Yes

Additional Feedback: === After rebuttal === I have read the response. I hope that the authors will polish their writing in the analysis as their promise.


Review 4

Summary and Contributions: This work studies the expressiveness of two-layer (mostly RELU) networks, focusing on the question of how many hidden neurons are sufficient to interpolate the training data and the optimality of the construction in terms of weight size. The first contribution is to extend the result of Baum, 1988 from binary to arbitrary labels: n/d neurons is sufficient to fit arbitrary labels. Then, the paper extends this result to NTK and harmonic networks.

Strengths: I have some concerns about the correctness below. But if they are addressed, the results on expressiveness of NTK and harmonic networks are new and interesting.

Weaknesses: The “wide-spread” assumption (i.e., data points cannot be too close) is restrictive. To my knowledge, most practical datasets do not satisfy this, for example MNIST.

Correctness: Some important proof steps are incomplete (proof of Lemma 1, proof of the extension of Proposition 3...), so I have concerns about the correctness. I give more details in the additional feedback section. That said, it could be that I missed something, so I am happy to be proven wrong and willing to change my stance if my concerns are properly addressed. I do not think the term “memorization”, which is used all over the paper, is a right terminology. I think it should be reserved for a learning algorithm rather than a construction. The term “interpolation” would be appropriate.

Clarity: My feeling is the paper is hard to follow at times. I also find it hard to distinguish the main contributions from existing results. The introduction is filled with theoretical statements and proofs that I believe should be put to the main technical section. Introduction should probably focus on high-level ideas and a summary of the results (by informal statement and discussion) in a clear distinction with related work. In my opinion, Section 2 should be moved to an earlier place. I am not sure Theorem 1 is an informal version of which formal one.

Relation to Prior Work: Yes

Reproducibility: Yes

Additional Feedback: Clarification questions: 1. In Theorem 1, what does the first inequality say? If there is any i such that the square loss is > 1, then it must not hold for an arbitrarily small epsilon. 2. I do not follow how to construct g in the proof of Lemma 1 (Line 69-70). Line 86, the sign of b_l is not consistent with Eq. 1. Expanding the inline equality (at the end of the same line) would be helpful for the reader. 3. What is f_{u, v, b} in Line 180? Is it defined in Equation (3). Moreover, (3) is not mathematically precise, shouldn’t LHS of (3) \lim_{\delta -> 0}? 4. Can the authors explain why it is possible to assume f_{u, v, b}(x_i) \in F_2(RELU) in Line 184? 5. Is there any intuition why the dependence on 1/\gamma is only logarithmic? Minor comments: Line 157, for a fixed training set X, why do we have randomness in H^\infty? Bibliography style does not follow NeurIPS format. ========== AFTER REBUTTAL =========== The authors have addressed my concerns and clarified some confusion. I increase my score accordingly.

[Author Response · NeurIPS 2020]

We thank the reviewers for their detailed reports, and we are particularly happy that they appreciated one of our key
innovation to use complex weights to cancel higher order moments. We view this as the most original part of our work.

**Reviewer 1:** *1.* We agree with you that it is important yet non-obvious that our set of weights can be found by SGD
/GD. We do believe that with more work one could reproduce our results directly with SGD/GD (see line 118). The
paper being already quite long, we decided to postpone these technicalities to future extensions of our work.
*2.* We meant this informally, in the sense of combining iteratively weak learners (namely, our procedures add either a
single or a couple of neurons at each iteration).
*3.* From the point of view of the proofs, the multiplicative error seemed more natural. For the Baum network the
difference is immaterial (since one achieves exact fit), and for the NTK network it is only a logarithmic difference. For
the Harmonic network however the issue is more severe, and it comes back to the difficulty we mention on line 77. It is
a great open problem in our opinion to fix this issue for the Harmonic network.
*4.* We apologize for the brevity of Lemma 1's proof, which we viewed as a mere technicality. We did not expand on
it because the argument is standard in the convex optimization literature, but we will add further details in the final
version. Also note that the proof of Theorem 6 contains essentially the same argument, in a detailed form.
*5.* Thank you!
*6.* The key difference with Ji and Telgarsky 2020 is that the latter work makes more assumptions on the data, and in
turn it allows them to construct smaller networks. We believe the two papers are complementary.
*7.* The non-polynomial assumption is necessary to invoke universality theorems (line 160).
*8.* We agree with you that more precise definitions of the NTK regime would be useful, we will add them in the revision.
*9.* Basically we have to assume that $n$ is not exponentially large in the dimension (to ensure that the data is well-spread),
and the parameter $m$ is connected to the exponent relating $n$ and $d$.
*10.* It is not optimal only in the sense that the dependency on epsilon can be improved. It is the same open problem as
the one mentioned in point 3 above.

**Reviewer 2:** Thank you for the positive feedback! Following your suggestion we will add further comments on
Daniely's work on and [Bresler and Nagaraj 2020] (see also response to Reviewer 3, in particular BN2020 requires
much larger networks than ours).

**Reviewer 3:** Thank you for thoroughly reading the paper and for mentioning [Bresler and Nagaraj 2020]; we plan to
add a detailed discussion comparing to this work, which we summarize here: One notable similarity is in fact that both
works rely on an iterative procedure. However, a crucial difference between the two memorization results is that the
result of [BN20] applies for the *random features* regime whereas in our work the iterative procedure actually updates
the $w_j$'s, and a much smaller number of neurons is needed as a result. The similarity in the use of complex weights
seems to be superficial, and is done in a completely different context.

We will make a careful pass on the paper to fix the issues raised by the reviewer, as detailed below:
*Line 415*: We apologize for this mix-up, which is a result of a last-minute change of notation: $\alpha$ should be replaced by
$b$, hence $\langle c, u \rangle = \alpha$. Other than that, we believe that the argument is correct.
*Line 540*: The constant $C_m$ comes from analysis of polynomial functions in Gaussian space. Since the parameter $m$
relates between $\gamma$ and $n$ there is an indirect dependence of $C_m$ on the those quantities, but thanks to Assumption (22)
everything can be expressed in terms of $m$. The constant is actually super-exponential in $m$ and we will make it explicit
in the final version.
*Line 175*: By adding an additional neuron we can assume that the label 1 is the minority. Hence $\lceil \frac{n}{2d} \rceil$ iterations suffice
to cover all relevant labels.

**Reviewer 4:** We thank you for your detailed reading of the paper! We will take into consideration your suggestion
for the organization of the paper, and focus the rebuttal mostly on correctness. First, regarding the weakness, note
that the results in Section 3 do not require any assumption on the data (besides full rank, which is essentially always
the case). We agree that in Section 4 and 5 the "wide-spread" assumption exclude certain real datasets, and it is a
great open problem on how to go beyond this assumption. We felt however that it was a natural first step to analyze
iterative methods such as the ones proposed in Sections 4,5. Secondly, regarding your memorization comment: Our
constructions are actual learning algorithms, in the sense that they take as input the training data, and output a set of
weights. It is not however a classical GD/SGD type learning algorithm; it would be a very nice extension to our work to
prove similar results in this case. Now regarding your clarificaiton points:

*1.* Note the normalization $1/n$. The point is that an error of any fixed (but small) value of $\varepsilon$ can be attained.
*2.* We will provide more details in the final version. In the meantime, the proof of Theorem 6 contains essentially the
same iterative procedure (in fact, a generalization of Lemma 1).
*3.* It should be $f_{u,v,b}$ in (3). Thank you for catching this! Also note the line just above (3), which agrees with $\delta \to 0$.
*4.* Since ReLU is piece-wise linear, we can take $\delta$ small enough to ensure that (3) holds true even without the limit.
*5.* Note that albeit the log function, the expression diverges at $\gamma = 1$ so this dependence is actually polynomial.

[Meta-Review · NeurIPS 2020]

The paper was reviewed by experts on the topic and discussed after authors rebuttal. Results were found to be interesting and valuable. The reviewers comments should be taken into account while preparing the final version of the paper.